# Layerwise Learning Rates for Object Features in Unsupervised and Supervised Neural Networks And Consequent Predictions for the Infant Visual System

## Abstract

To understand how the brain and mind develop in infancy, it is necessary to develop testable computational models. In recent years, DNNs have proven valuable as models of the adult brain. In the domain of object recognition, for example, convolutional DNNs currently provide the best way to predict the response to a novel stimulus in the ventral visual stream of the adult monkey and adult human. Given this success in modelling the mature brain, we propose them as candidate models for the learning process. We tested if learning in different DNNs yields distinct trajectories of development at the large scale that could be testable in future brain imaging experiments. First, we quantified the development of explicit object representation at each layer in the network at various points in the learning process, by freezing the convolutional weights and training an additional linear decoding layer. In two DNNs trained in a supervised way (CORnet-S and Alexnet), object representation was strongest in the top layer throughout learning. Infants, however, have only extremely impoverished access to labels, and so unsupervised learning is likely a better model for their learning process. Infants are known to identify clusters of similar-looking objects, and so we evaluated an unsupervised DNN with a clustering objective (DeepCluster). Through learning, there was a bottom-up sweep in the development of object representations, with the penultimate layer strongest at the end. These results show that different training strategies yield different layerwise trajectories of development, which could be empirically measured in infants with neuroimaging. In all three DNNs, we also tested for a relationship in the order in which infants and machines acquire visual classes. We found different patterns, which could be driven by the different visual features that DNNs and humans exploit, or by differences in the objective function used for learning. Our results show that DNNs can make distinct and testable predictions for infant development. The parallel that we are creating highlights ways in which knowledge of the infant brain and mind could guide the design of new DNNs, such as by providing insight into the selection of auxiliary tasks for unsupervised training, from the myriad available.

## 1 Introduction

### 1.1 The Need for Models of Development in Infants

Humans are helpless for a long time during infancy when compared with most other species. During this helpless period, the brain and mind develop as a result of genetically programmed maturation in concert with experience from the environment. One way to measure development during the helpless period is by examining infants' capabilities through behaviour. However, despite great ingenuity in experimental design, due to the limited behavioural repertoire of infants, it has proven difficult to obtain more than a basic understanding of how the brain and mind are developing. Furthermore, some brain systems may undergo extended development prior to any manifestation in behaviour (Tolman & Honzik, 1930). Therefore, in addition to measuring behaviour a new and promising

complementary tool is magnetic resonance imaging (MRI), which can measure the structure, wiring and function of the infant brain.

To understand why infants' brains and minds change in the way they do, we need to create models of the mechanisms of development. This is critical if we are to understand how development is disrupted in infants with brain injury. Infants that are born very preterm, or that have potentially negative event during birth are admitted to the neonatal intensive care unit (NICU). At present, a significant proportion of NICU infants develop cognitive, behavioural or social impairments later in life. To detect these impairments earlier, and to design effective targeted interventions, we need to create models of brain and mind development in young infants. As it is difficult to intuitively imagine what the mind of a developing infant is like, it is necessary to create computational models of infant learning that can be tested against real data.

## 1.2 DNNs as a Model for Object Vision in Adults

A promising first brain system for which computational models of infant learning could be built is the object recognition system in the ventral visual stream. This system is relatively well understood in adults and there are proven neuroimaging methods that can be used to probe its representations and to test the hypotheses that will be generated by our computational models in due course. Indeed, in adults, there is substantial work on modelling the ventral visual stream. A key discovery from this work has been that the most straightforward strategy, of first collecting neural data and then creating models to fit it, has not been very effective (Yamins & DiCarlo, 2016). The issue is that it is only feasible to acquire a relatively small quantity of neuronal data, but the computational models capable of performing object vision have many millions of parameters. A more effective strategy, therefore, is to instead use computational models that are optimised to perform object vision to high accuracy, and then to use these as models of neural activity (Yamins & DiCarlo, 2016). Using this strategy, it has been shown that DNNs designed for the ImageNet Large Scale Visual Recognition Challenge (ILSVRC) can predict patterns of neural activity in the ventral visual stream, as measured with functional magnetic resonance imaging (fMRI) (Khaligh-Razavi & Kriegeskorte, 2014; Güçlü & van Gerven; Wen et al., 2018), electroencephalography (EEG) and behavioural studies (Cichy et al., 2016). It is important to note that the claim in these studies is not that there is a one-to-one mapping between artificial neurons and biological neurons. Rather, it is that some of the macroscopic aspects of the activity patterns in the DNNs and the brain are similar. For example, the ventral visual stream forms a hierarchy, with visual input from the eyes being passed forwards through a number of brain structures, and gradually being transformed from a representation of the visual input into a representation of semantic category. A quantitatively similar hierarchy of representations is found through the layers of the DNN (Güçlü & van Gerven; Wen et al., 2018).

## 1.3 Infants are Learning, Do They Learn Like DNNs?

Some aspects of biological object vision are innate, such as a preference for looking at faces, which is apparent in the first hour after birth (Johnson et al., 1991). However, much must be learned through visual experience, as the genetic code alone is limited in capacity (estimated to be 150-750 MB[1]) and would be substantially exhausted by the parameters of deep learning models (50-150 MB (Bianco et al., 2018)). Even if sufficient storage were available, many objects relevant to infants today (e.g., baby bottle, or a Pokémon (Janini & Konkle, 2019)) were invented too recently for innate codes to have evolved for them. A great deal of object vision knowledge, therefore, is learned.

There is evidence that much of this learning is already happening in early infancy. Infants aged 3-4 months old can identify statistical regularities in sequences of pictures (Quinn et al., 1993; French et al., 2004). And, by 6 months, infants start to look at a visual class corresponding to a concurrently spoken label (Bergelson & Swingley, 2012)[2]. In adults, fMRI of the ventral visual stream has found that there are regions that are selective for particular visual classes, such as faces, body parts or places (Kanwisher et al., 1997; Epstein & Kanwisher, 1998; Downing et al., 2001). In 4-6 month old infants, at least partial selectivity is already present in the ventral visual stream (Deen et al., 2017), although it continues to develop for many years (Gomez et al., 2017).

---

[1] https://www.quora.com/How-many-bytes-memory-size-is-a-humans-DNA

[2] although vocabulary remains very limited until after the first birthday, when it typically begins to grow rapidly.

DNNs have provided a useful model of adult object vision. As these DNNs learn from visual "experience" they are therefore candidate models of infant learning. We do not expect them to capture the precise details of the learning process as, for example, the learning rule of back propagation is not biologically plausible (although biologically plausible mechanisms can approximate it (Richards et al., 2019)). Rather, our overarching hypothesis is that DNNs might capture some of the broad macro-scale characteristics of learning.

## 1.4 AIMS OF THE CURRENT STUDY

In this work, we characterise two macro-scale properties of a number of computational models, with the aim of generating testable measurements for a future large-scale neuroimaging study.

1. Should we expect representations in brain regions of the visual hierarchy to develop simultaneously or asynchronously? An established principle of infant development is that brain regions underlying simpler functions develop first, and are followed by those underlying more complex functions (Charles A. Nelson in Shonkoff & Phillips (2000))[3]. To provide an initial prediction of how this might happen in the visual hierarchy we examined how representations developed in different layers of DNNs during training.

2. Are visual classes that are learned earlier by infants also learned earlier by DNNs? In infants, the acquisition of visual classes can be estimated by the onset of the receptive or expressive use of the words for the classes. It has been found that, when measured this way, some visual classes are learned before others, e.g., body parts and vehicles precede food and clothing (Braginsky et al., 2015). The link between the acquisition of words and brain representations in the ventral visual stream is supported by fMRI results examining the visual representations of mental imagery have shown that object classes can be decoded from brain representations when participants are tasked with reading a concrete noun (Anderson et al., 2015; Kellenbach et al., 2000). We ask whether some of this ordering is attributable to visual complexity, as reflected in the DNN classification performance.

## 2 METHODS

### 2.1 CHOICE OF NETWORKS AND TRAINING

#### 2.1.1 CORNET-S WITH SUPERVISED TRAINING

To date, studies which have shown a parallel between DNNs and the adult brain have used networks trained in a supervised manner (Khaligh-Razavi & Kriegeskorte, 2014; Güçlü & van Gerven; Cichy et al., 2016; Wen et al., 2018; Jozwik et al., 2018). To remain consistent with these studies, we started with a supervised network. Specifically, we used CORnet-S (Kubilius et al., 2018), as this was especially designed to meet the Brain-Score benchmark (Schrimpf et al., 2018) by capturing the architectural principles and showing strong predictivity of neural data from the ventral visual stream, while still achieving good classification performance. It has four layers mapping onto regions in the ventral visual stream (V1, V2, V4 and IT).

#### 2.1.2 DEEPCLUSTER WITH UNSUPERVISED TRAINING

Infants' access to labels is extremely impoverished, making supervised learning an unlikely training curriculum for infant learning. In order to develop computational models that capture the dynamics of infant learning it is necessary to evaluate unsupervised training. One current state-of-the-art unsupervised strategy for learning visual features for object recognition is DeepCluster (Caron et al., 2018), which adjusts convolutional weights to create clusters of images that yield similar patterns of activation. This has a parallel with behavioural results of infant learning, in that they are known to create clusters of similar images, and to be sensitive to deviants from that cluster (Quinn et al., 1993; French et al., 2004). DeepCluster uses a simple but elegant technique for self-supervised learning, in which the start of an epoch sees each image (from ImageNet) being passed forward through a convolutional network, and the resulting output activations are then clustered across all of

---

[3]This influential work has been cited more than 7000 times.

the images using k-means. These clusters are assigned labels, which are subsequently used to learn the weights of the convolutional network with stochastic gradient descent on batches of images in the typical way.

As in Caron et al. (2018) we used 10,000 clusters. We used AlexNet (Krizhevsky et al., 2012) as the convolutional network, containing five layers each with a single convolutional layer. These layers were modified as in Caron et al. (2018) with the local response normalisation layers removed and batch normalisation used instead (Ioffe & Szegedy, 2015). Also like Caron et al. (2018), a fixed linear transformation based on Sobel filters was used on the input to remove colour and increase local contrast.

### 2.1.3 ALEXNET WITH SUPERVISED TRAINING

CORnet was trained in a supervised way, and DeepCluster in an unsupervised way. Any difference in the results might be due to this difference in supervision. However, the convolutional networks at the heart of these two networks also differ, which could cause further differences. To control for this, we therefore repeated the experiments using the same AlexNet variant as in DeepCluster, but trained in a supervised way.

### 2.2 LEARNING TRAJECTORIES

The value for object recognition of the representations was assessed for each of the layers in the networks (4 layers for CORnet, and 5 layers for DeepCluster and Alexnet). Specifically, we quantified the explicit representation (DiCarlo & Cox, 2007) of object class using the method by Zhang et al. (2017) of freezing weights in the convolutional layers, and training a linear decoder on the output of each layer to decode the ImageNet categories. This was done across epochs in the learning process, to capture the development of the representations in each of the layers.

### 2.3 SUMMARISING LEARNING OF VISUAL CLASSES

To summarise the learning curves of the individual classes we fitted the performance with the curve

$$p = A(1 - exp(-kt)) \tag{1}$$

where p is top-5 precision, t the epoch, A the asymptotic level of performance, and k the learning rate. Fitting minimised least squares with the addition to the cost function of two regularisation terms equal to $k^2$ (to discourage implausibly high learning rates) and $(A < 100) * (A - 100)^4$ to discourage A values greater than 100%.

### 2.4 AGE OF ACQUISITION IN INFANTS

We also compared the order in which visual classes are acquired in infants and machines. Unfortunately, there is no data available at present of the order in which infants acquire different visual classes. Therefore, we used a proxy, which is the estimate of the age of acquisition (AoA) of the word for the class (Kuperman et al., 2012). While a number of linguistic factors are known to affect when words are first used, including the frequency of the word in language and its number of phonemes, critically the second strongest factor is the "concreteness" of the word (Braginsky et al., 2015). Concreteness is the degree to which a word is associated with a perceptual representation, and is typically obtained by asking people to rate it. The concept has been experimentally validated in many studies, and concrete words have been shown to more readily evoke visual representations in the brain which are capable of being decoded as visual classes using fMRI and EEG (Kellenbach et al., 2000; Anderson et al., 2015). Thus, the association of AoA with concreteness suggests that the strength of the visual representation of a class may have an effect on when its label is acquired. This might happen due to one clear constraint; a child cannot name a visual class before the representation of that visual class has been developed. So, we tested if the visual classes that are labelled earlier in infants are learned more quickly by DNNs.

Using the Natural Language Toolkit (NLTK), the WordNet synsets for the 1000 ImageNet classes were compared to Kuperman et al. (2012) database of 30,000 English words with AoA ratings using Leacock et al. (1998) semantic similarity metric. The classes with the highest similarity score were

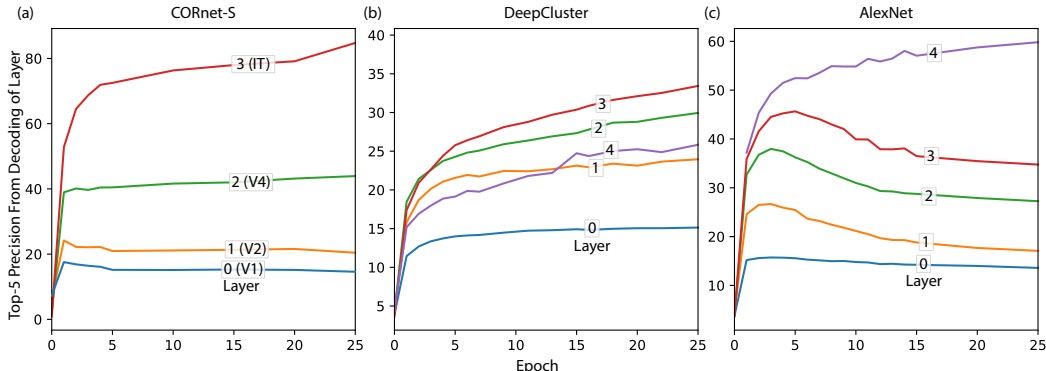

Figure 1: Explicit representation of visual class measured in the three networks during training, as measured by top-5 precision. The corresponding cross-entropy loss is shown in Appendix B, Fig. 5.

considered as matching, and manually inspected for any incorrect comparisons or synset definitions. These were deleted, leaving a total of 308 classes on which further analyses were conducted.

### 2.5 CLASS CATEGORISATION

To provide a visualisation of when different types of classes were learned, we clustered the 308 classes using Leacock et al. (1998)'s metric and then clustering (scipy.cluster.hierarchy.fcluster) to yield 20 classes. By visual inspection, we then attached a label to each of the class clusters.

### 2.6 IMPLEMENTATION

Training was run on AWS using the Deep Learning AMI version 24.0 on either a p2.8xlarge instance (8 x NVIDIA K80 GPUs with 488 GB of RAM) or a p3.8xlarge instance type (4 NVIDIA Tesla V100 GPUs and 244 GB RAM), using Python 3.6 with Pytorch 1.1. Spot instances were used to reduce cost. The three networks were trained from scratch. The ISLVRC 2012 set was used for training and validation. The CORnet-S code was obtained from `https://github.com/dicarlolab/CORnet`. DeepCluster, Alexnet and the linear classifier implementation were from `https://github.com/facebookresearch/DeepCluster`.

## 3 RESULTS

### 3.1 AIM 1: HIERARCHICAL DEVELOPMENT OF REPRESENTATIONS IN DIFFERENT LAYERS DURING TRAINING

#### 3.1.1 CORNET

Explicit representation of object class in the four layers of the CORnet network during training is shown in Fig. 1a. The earlier layers in the hierarchy (V1, V2 and V4) reached their asymptotic level quickly (around epoch 1), but IT continued to learn until at least epoch 25. However, although IT took longer to reach its asymptote, even after minimal training (epoch 1) it contained greater explicit information than the lower layers.

In human infants, such a learning scheme would therefore be be indicated by earlier maturation of lower-order visual processing regions (V1, V2 and V4), than higher-order brain regions (e.g., IT) in the developing brain. However, even early in development, infant IT would be expected to contain stronger explicit representations of object class as was seen for the IT layer of CORnet. It may be thought that this result is inherent to how IT is built to function; it was optimised to create explicit representations of objects, therefore this signature is found throughout training. However, when examining the results from later models we find that this is not the case.

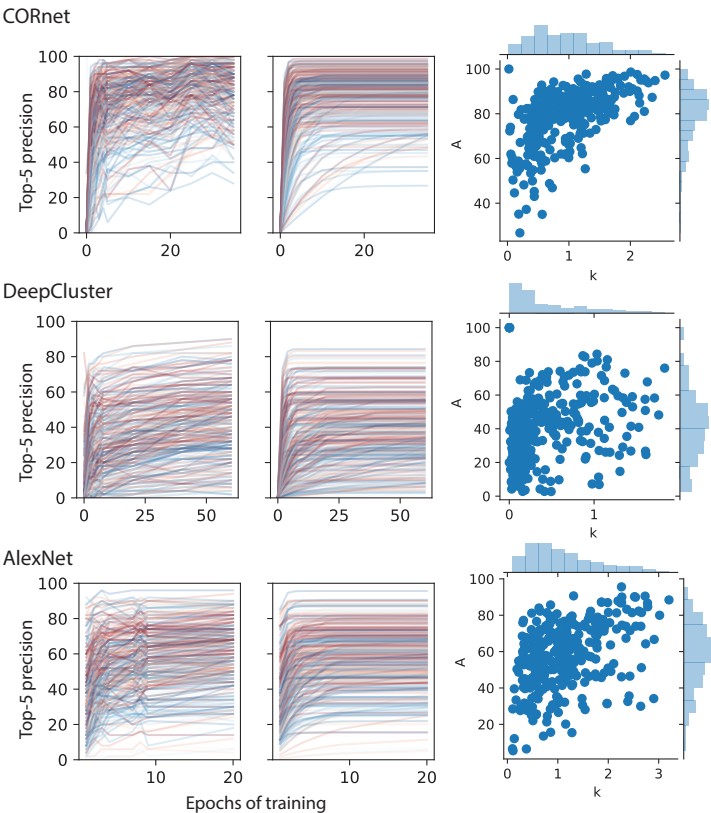

Figure 2: Left: Precision for each visual class during training in the most informative layer of the network - the top layer for CORnet and AlexNet, and the penultimate layer for DeepCluster. The colour of the curves shows AoA of the class's name for infants (blue to red for low to high AoA). Centre: The learning curves were parameterised with a fit (Eqn. 1), shown here. Right: Distributions of the fit parameters. They were correlated for all networks (CORnet $r(308)=0.62$, $p<0.001$; DeepCluster $r(308)=0.36$, $p<0.01$; AlexNet $r(308)=0.39$, $p<0.001$), showing that classes that were learned more quickly were also converging on a higher asymptote.

### 3.1.2 DEEPCLUSTER

In contrast, for DeepCluster, representations mature in a more "bottom-up" manner (Fig. 1b). Specifically, the explicit representation of object class does not monotonically increase with layer - even at the end of 60 training epochs, layer 3 contains stronger representations of object class than layer 4. Furthermore, the order of the layers varies through training, with layer 2 stronger than layer 3 early in training, and layer 1 containing stronger representations of object class than layer 4.

Extrapolating to the brain, this more developmentally plausible unsupervised strategy would be indicated not only by higher-order regions in the ventral visual stream developing more slowly, but by earlier regions initially leading in the presence of representations of object class. This hypothesis is testable in future fMRI experiments.

In supervised training, object labels are provided at the top layer of the network, and so it is perhaps not surprising that even at early epochs the entire network, including the upper layers, are maturing. This contrasts to unsupervised learning during which the only source of information is the visual input, and so it is perhaps not surprising that maturation proceeds in a more bottom up manner; until good representations have developed in the early layers, there is poorer information at higher layers. This is consistent with the simple-to-complex maturation theory (Charles A. Nelson in Shonkoff & Phillips (2000)).

### 3.1.3 ALEXNET

CORnet and DeepCluster are not just different in their training strategies, but also in the convolutional networks at their heart. To control for this, we repeated training with AlexNet. The results in Fig. 1c show that even when the same convolutional network as DeepCluster is used, but instead with a supervised training strategy, the bottom-up learning trajectories of DeepCluster are eliminated.

## 3.2 AIM 2: ACQUISITION OF VISUAL CLASSES

Strikingly, explicit object representation in the lower layers of AlexNet actually reduced from epochs 3-5 onwards when assessed with top-5 or loss (Appendix B, Fig. 5c), an effect which was also seen in the supervised CORnet albeit much more weakly. In DeepCluster, this reduction in explicit representation did not appear, suggesting that it may in fact be a feature of supervised learning.

In the second aim, we compared machine and human learning across visual classes. The learning curves were fit well by the model (Eqn. 1, left two columns of Fig. 2). The joint distribution (Fig. 2, right column) showed that classes which were learned quickest were ultimately learned best, as the two fit parameters were strongly correlated for all three models (CORnet $r(308)=0.62$; $p<0.001$; DeepCluster $r(308)=0.36$, $p<0.01$; AlexNet $r(308)=0.39$, $p<0.001$)

These fit parameters were then used to compare the machine with human learning. Paradoxically, classes learned more precisely by the model were if anything learned *later* by infants (Fig. 3, correlation of AoA and parameter A, CORnet $r(308)=0.11$ $p=0.06$; DeepCluster $r(308)=0.10$, $p=0.09$ ; AlexNet $r(308)=0.14$, $p<0.02$). Although classes which were learned more precisely were in general learned more quickly in the model, there was no relationship observed between learning rate parameter (k) and infant AoA.

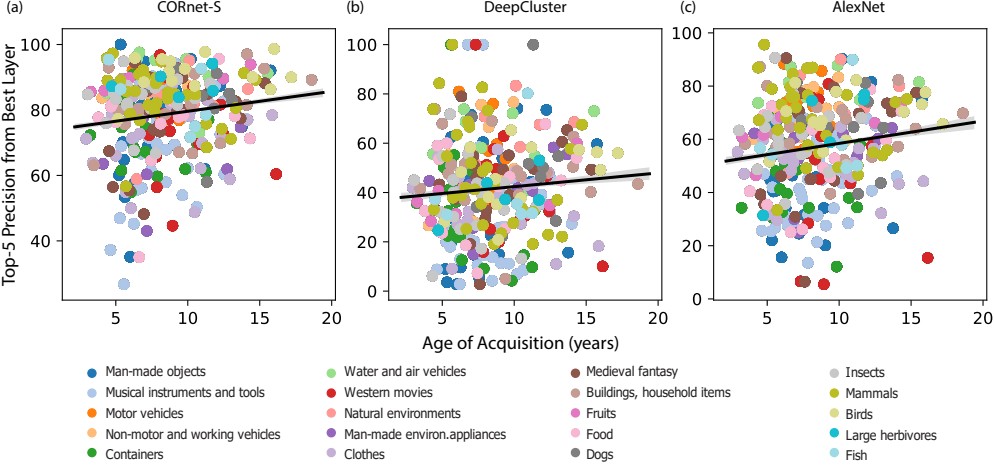

Figure 3: Relationship between infant age of acquisition and asymptote of machine performance for each visual class in the most informative layer of each network. The colours show different semantic groupings.

## 4 IMPLICATIONS, LIMITATIONS AND FUTURE WORK

### 4.1 POTENTIAL OF DNNs TO PROVIDE INSIGHT INTO INFANT DEVELOPMENT

In Aim 1, the macroscale distribution of object representations was found to be affected by learning strategy, with supervised training concentrating object representations near the output layer, while unsupervised training yielded distributed representations across layers, peaking in the penultimate layer. Learning strategy also led to layerwise differences in the order of learning, with unsupervised learning showing a bottom-up sweep in the object representation, but supervised learning an

unchanging order with the top layer leading throughout. A control experiment showed that these effects were due to the learning strategy rather than the DNN architecture. In future work, it will be important to extend the investigations to further DNNs. These could test generalisation to variants on the strategies, such as local aggregation, a recently proposed unsupervised training objective, which like DeepCluster learns a visual embedding based on clustering of images (Zhuang et al., 2019). It would be informative to investigate a wider range of objectives, for example testing DNNs that exploit other structures in visual input such as temporal prediction (Lotter et al., 2016) or cross-modal learning (Wang et al., 2013). Future work could also investigate the benefit of making DNNs more similar to brains at the implementation level, for example by using an alternative to batch normalisation applicable to online learning[4]. These macroscale differences in layerwise learning could then be tested in future infant neuroimaging, to identify which models best approximate human learning.

In Aim 2, we found only a counter-intuitive relationship between the visual classes that are learned best by DNNs and those that are named first by infants. This might be because of the limitations of this measure in infants. The AoA of a label is only an approximation to when a visual class is acquired, as it is affected by other properties including the number of phonemes in the label. In future work, we will measure when infants acquire visual categories. There are other factors that influence when infants learn to identify a visual class in addition to the class's visual "accessibility". One is the frequency of the class in the environment. The ImageNet classes are esoteric and unecological, with a high preponderance of dog breeds, for example. More human-like learning will probably require more human-like (or baby-like) training sets. Another factor is the innate or learned reward value of the class to the baby (e.g., the mother's face or their milk bottle may be learned earlier).

## 4.2 POTENTIAL INSIGHTS FROM INFANT DEVELOPMENT TO GUIDE THE DESIGN OF DNNS

The current state-of-the-art DNNs for visual recognition use supervised training, but there is strong interest in developing unsupervised strategies, as unlabelled data is more plentiful and cheaper. Diverse unsupervised strategies have been demonstrated, such as cross-channel colour prediction (Zhang et al., 2017) and feature counting (Noroozi et al., 2017). Given the enormous space of possible unsupervised learning strategies, it is unclear how to direct investigations in a principled way. Knowledge from the human visual system such as the nature of motion and colour representations and how they develop in infancy could guide this search, and the layerwise trajectories of development could provide a constraint for biomimetic DNNs.

There is recent evidence that DNNs use textures rather than shape to recognise objects, while it is the reverse in humans (Geirhos et al., 2018a). The specific features relied upon by DNNs is thought to underlie the lack of robustness to noise, as is evident in adversarial attacks (Geirhos et al., 2018b). This difference in feature weighting might be one cause of the different trajectories of learning for different visual classes between machine and human found in Aim 2. Assessing DNNs in how similar their acquisition order is to humans might therefore provide a valuable benchmark to guide the development of more noise-robust DNNs.

## 4.3 CONCLUSIONS

DNNs were inspired by the brain. Although DNNs learn like humans from large quantities of data, there is little work to build formal connections between infant and machine learning. Such connections have the potential to bring considerable insight to both fields but the challenge is to find defining characteristics that can be measured in both systems. This paper has addressed this challenge by measuring two characteristic features in DNNs that can be measured in infants.

### 4.3.1 ACKNOWLEDGMENTS

---

[4]https://cbmm.mit.edu/sites/default/files/publications/CBMM-Memo-057.pdf

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

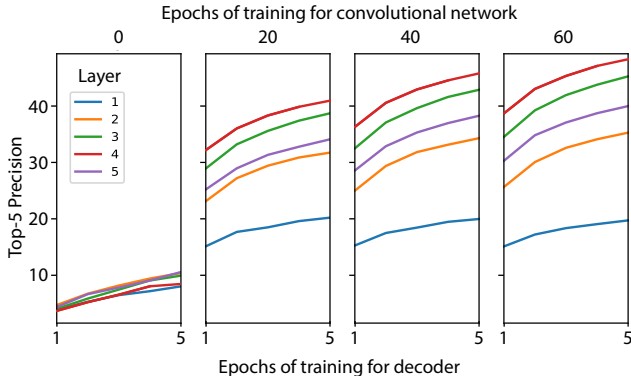

Figure 4: Top-5 precision as a function of training epochs of the top decoding layer for DeepCluster.

## A  APPENDIX: DETERMINING NUMBER OF TRAINING EPOCHS FOR THE OBJECT DECODER

Training the object decoders was the most computationally expensive part of this project, as one was trained for every layer across many epochs and models. It was therefore necessary to use as few training epochs as possible. To evaluate how many were needed, we trained decoders for 5 epochs on features from a sample of convolutional training epochs (0, 20, 40, 60) and all layers (Fig. 4). It was found that while there was a steady increase in decoding performance up to (and presumably beyond) the 5 epochs, the relative performance across different layers, or epochs, was broadly captured by epoch 2. For further analyses we therefore used 2 epochs of training for the decoding layer.

## B  APPENDIX: FURTHER MEASURES OF CHANGES IN OBJECT REPRESENTATION THROUGH LEARNING

Fig. 1 showed the layerwise changes in top-5 precision through learning. Fig. 5 shows the corresponding changes in cross-entropy loss.

As DeepCluster learned more slowly than the supervised networks, we extended training to 70 epochs (Fig. 6). It can be seen that it was continuing to learn, particularly in the higher layers, but the order of the layers did not change within this range.

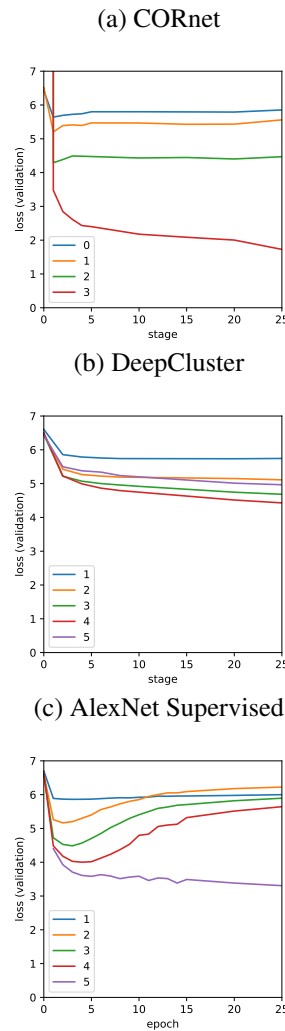

Figure 5: Cross-entropy loss during training for each of the three networks, illustrating the change in explicit representation of visual classes.

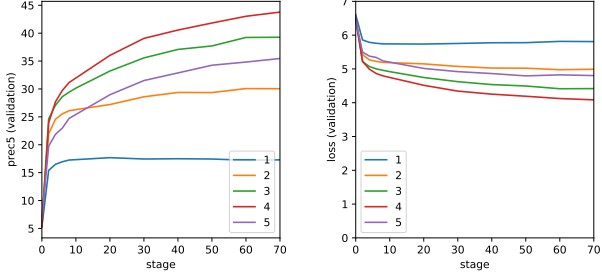

Figure 6: The unsupervised network, DeepCluster, learned more slowly than the supervised networks (Fig. 5). To test if the layers with the strongest explicit object representation changed over a longer period of extended learning, we trained the convolutional layers to 70 epochs.

