# OpenReview forum: "Layerwise Learning Rates for Object Features in Unsupervised and Supervised Neural Networks And Consequent Predictions for the Infant Visual System"
_ICLR.cc/2020/Conference — Reject_

### Official Review · AnonReviewer1 · 2019-10-23
**Official Blind Review #2**

**Rating:** 3

**Review:**

Summary

The paper aims to understand how object vision develops in infancy and childhood by using deep learning models. In particular, it chooses two deep nets, CORnet and DeepCluster to measure learning. CORnet is supervised and is designed to mimic the architecture and representational geometry of the visual system. It tries to quantify the development of explicit object representations at each level of this network through training by freezing the convolutional layers and training an additional linear decoding layer. The paper evaluates the decoding accuracy on the whole ImageNet validation set for individual visual classes. DeepCluster differ in both supervision and in the convolutional networks. To isolate the effect of supervision, it ran a control experiment in which the convolutional network from DeepCluster (an AlexNet variant) is trained in a supervised manner. The paper tries to draw conclusions on how learning should develop across brain regions in infants. In all the networks, it also tested for a relationship in the order in which infants and machines acquire visual classes, and found only evidence for a counter-intuitive relationship.

Limitations

The topic is extremely interesting and worth intense study. However, the approach is not convincing. CORnet may have some relevances. It is not clear how well it models the representational geometry of the visual system. It is even less clear whether DeepCluster is relevant. Why would it be related to infant learning?

The whole idea of using DNN to infer biological learning is built on shaky ground given how little we know the learning mechanism of the brain. In particular, back propagation is not widely considered possible in biology. Given the learning mechanism may be very different. What is the basis of using DNN to study the infant learning?

The findings are also not very surprising and offer much for the community.

Given the paper lacks rigor and findings, it does not meet the bar of ICLR.

**Experience Assessment:**

I have read many papers in this area.

**Review Assessment: Checking Correctness Of Derivations And Theory:**

N/A

**Review Assessment: Checking Correctness Of Experiments:**

I carefully checked the experiments.

**Review Assessment: Thoroughness In Paper Reading:**

I read the paper thoroughly.

---

> ### Author Response · Authors · 2019-11-13
> **Revisions to address your comments**
>
> Thank you for taking the time to consider this paper and for providing constructive suggestions.
>
> To articulate the rationale for using DNNs as a model of human learning, we have thoroughly revised the introduction. We address your comment on backpropagation in three ways:
> * Although backpropagation is not biologically plausible, it can be approximated by a number of biologically plausible mechanisms, including the synaptic plasticity observed in top-down processes (Richards et al., 2019; Sinz et al., 2019). This has been clarified in section 1.3.
> * The parallel we intended to draw was unclear. DNNs are currently the best models of the adult visual system, but they do not attempt to map artificial neurons onto biological ones. Instead, activation profiles across the layer of a DNN show macroscale similarities to activation patterns in a brain region. For example, the degree to which two images cause similar patterns of activity in a DNN has proven predictive of the degree to which they create similar patterns of activity in the brain, when a person or monkey is viewing them. We propose that similar macroscale statistics might provide insight into learning in the infant brain. This has been clarified throughout.
> * We have cited two recent high-profile reviews presenting the value of the DNNs as a model of the brain (Richards et al., 2019, Sinz et al., 2019).
>
> We have expanded upon the motivation for choosing specific DNNs:
> * CORnet-S was chosen as it is currently the top performing DNN on the Brain-Score benchmark (Schrimpf et al. 2018; www.brain-score.org) and is therefore the best existing model of neural firing in the mature ventral visual stream.
> * Infants’ access to labels is extremely impoverished. However, there is evidence that by 3-4 months of age, they can cluster together visually similar stimuli (Quinn, 1993). DeepCluster was therefore chosen as it uses a clustering strategy, and it is one of the best performing unsupervised strategies for learning object representations. Again, it was not intention to claim a parallel between the specifics of the network and the brain, but rather that the DNN might capture some of the macroscale properties of the brain.
>
> Finally, we have thoroughly revised the discussion (section 4) to clarify the contribution of this work to neuroscience and machine learning.
>
> Thank you also for identifying the minor issues, which we have addressed. We hope you agree that the paper is greatly strengthened.
>
> References
>   Quinn PC, Eimas PD, Rosenkrantz SL. Evidence for representations of perceptually similar natural categories by 3-month-old and 4-month-old infants. Perception. 1993 Apr;22(4):463-75.
>   Richards BA, Lillicrap TP, Beaudoin P, Bengio Y, Bogacz R, Christensen A, Clopath C, Costa RP, de Berker A, Ganguli S, Gillon CJ. A deep learning framework for neuroscience. Nature neuroscience. 2019 Nov;22(11):1761-70.
>   Schrimpf M, Kubilius J, Hong H, Majaj NJ, Rajalingham R, Issa EB, Kar K, Bashivan P, Prescott-Roy J, Schmidt K, Yamins DL. Brain-Score: which artificial neural network for object recognition is most brain-like?. BioRxiv. 2018 Jan 1:407007.
>   Sinz FH, Pitkow X, Reimer J, Bethge M, Tolias AS. Engineering a less artificial intelligence. Neuron. 2019 Sep 25;103(6):967-79.

---

### Official Review · AnonReviewer3 · 2019-10-23
**Official Blind Review #3**

**Rating:** 3

**Review:**

This paper aims to examine whether DNNs are a good model of infant
behavior and development.  The paper is very well written and easy to
read.

The goal was to compare the development of object representations
across layers with the development in children and to compare the
order of learning of different object classes.

The work did show that unsupervised training results in a different
pattern of layer learning than supervised learning, but neither form
of learning was able to model the development in children.  Perhaps a
self-supervised multimodal learning system should be tried?


The decision to train the DeepCluster type net in a supervised way for
a control on training method vs architecture type is nice, but it would also
have been good to try other kinds of networks.


It is not clear that age of acquisition of the verbal word should be
related to age of acquisition of the visual concept.  The author's
state "A number of linguistic factors are known to affect when words
are first used, including the frequency of the word in language and
its number of phonemes, but the second strongest factor is the
"concreteness" of the word (Braginsky et al., 2015). This suggests
that the strength of the visual representation of a class has an
effect on when its label is acquired."   It is not clear to me how the concreteness
of a word relates to the strength of visual representation.

I don't think there is enough new insight gained from this paper for ICLR publication
at this stage.

Minor comments:

What are dashed lines in Figure 2 top left box?

Funding acknowledgement (especially with grant number) should not be
in an anonymous submission.

magenetic



**Experience Assessment:**

I have published in this field for several years.

**Review Assessment: Checking Correctness Of Derivations And Theory:**

N/A

**Review Assessment: Checking Correctness Of Experiments:**

I assessed the sensibility of the experiments.

**Review Assessment: Thoroughness In Paper Reading:**

N/A

---

> ### Author Response · Authors · 2019-11-13
> **Revisions to address your suggestions**
>
> Thank you for taking the time to consider our paper and for providing interesting and constructive feedback. We edited the paper accordingly, clarifying and correcting where appropriate.
>
> >> The work did show that unsupervised training results in a different pattern of layer learning than supervised learning, but neither form of learning was able to model the development in children.  Perhaps a self-supervised multimodal learning system should be tried?
> >> The decision to train the DeepCluster type net in a supervised way for a control on training method vs architecture type is nice, but it would also have been good to try other kinds of networks.
>
> We have now added the following paragraph to section 4.1.
> * “In future work, it will be important to extend the investigations to further DNNs. These could test generalisation to variants on the strategies, such as local aggregation, a recently proposed unsupervised training objective, which like DeepCluster learns a visual embedding based on clustering of images (Zhuang et al., 2019). It would be informative to investigate a wider range of objectives, for example testing DNNs that exploit other structures in visual input such as temporal prediction (Lotter at al., 2016) or cross-modal learning (Wang et al., 2013).”
>
>
> >> It is not clear that age of acquisition of the verbal word should be related to age of acquisition of the visual concept.  The author's state "A number of linguistic factors are known to affect when words are first used, including the frequency of the word in language and its number of phonemes, but the second strongest factor is the "concreteness" of the word (Braginsky et al., 2015). This suggests that the strength of the visual representation of a class has an effect on when its label is acquired."   It is not clear to me how the concreteness of a word relates to the strength of visual representation.
>
> We have added some further resources which we believe better support our rationale for using age of acquisition (AoA) as a proxy for visual class acquisition.
> * We have further unpacked the pros and cons of this measure. At present, there are no measures of the age of visual classes on a sufficient scale, which is now highlighted in section 2.4.
> * The relationship of concreteness to ventral visual stream representations has been established. Studies of “embodiment theory” using fMRI and EEG have shown that reading a concrete noun evokes visual representations of the object (Anderson et al. 2015, Kellenbach et al. 2000). This is now discussed in Aim 2 of the paper and section 2.4. These results suggest that learning a word will build upon these visual representations.
> Given the evidence and constraints, we believe that AoA is a valuable starting point.
>
> We hope you agree that the paper is greatly strengthened.
>
> References
>   Anderson AJ, Bruni E, Lopopolo A, Poesio M, Baroni M. Reading visually embodied meaning from the brain: Visually grounded computational models decode visual-object mental imagery induced by written text. NeuroImage. 2015 Oct 15;120:309-22.
>   Kellenbach ML, Wijers AA, Mulder G. Visual semantic features are activated during the processing of concrete words: Event-related potential evidence for perceptual semantic priming. Cognitive Brain Research. 2000 Sep 1;10(1-2):67-75.

---

### Official Review · AnonReviewer2 · 2019-10-24
**Official Blind Review #2**

**Rating:** 3

**Review:**

This paper attempts to model the development of the human visual system in infants, by training deep neural network architectures inspired by the human visual system on images from ImageNet, and learning a linear decoder on the outputs of each layer (following Zhang et al 2017) to measure how much information useful for distinguishing between classes is contained within each layer in the architecture. The paper measures the amount of class information in each layer over the progress of training.

I agree that deep networks could serve as good models for various parts of the brain, including the visual system especially given that convolutional networks have been inspired from studies of the visual system. However, the paper doesn't seem to provide any evidence for how the training process used for deep neural networks should correspond to the development of the visual system in infants. In particular, backpropagation is considered biologically implausible [1], whereas backpropagation serves as the main method for learning in the neural networks. Furthermore, neural networks have randomly initialized parameters, whereas it seems unlikely that human infants' brains would lack existing organization to such a drastic extent. In order for the results in this paper to hold greater weight, I would expect to see more evidence about how the neural network training process (also including aspects such as batch normalization, and the self-supervised clustering method in DeepCluster) are expected to correlate with learning in human brains.

For the above reasons, I vote to reject the paper. My conclusions above are based on my surface-level knowledge of neuroscience, so I welcome any clarifications or corrections from the authors about the above points.

[1] https://arxiv.org/abs/1502.04156

**Experience Assessment:**

I do not know much about this area.

**Review Assessment: Checking Correctness Of Derivations And Theory:**

N/A

**Review Assessment: Checking Correctness Of Experiments:**

I assessed the sensibility of the experiments.

**Review Assessment: Thoroughness In Paper Reading:**

N/A

---

> ### Author Response · Authors · 2019-11-13
> **Revisions to address your comments**
>
> Thank you for taking the time to consider our paper and for providing constructive feedback. We edited the paper accordingly, clarifying and correcting where appropriate.
>
> To better articulate the rationale for using DNNs as a model of infant visual development, we have thoroughly revised the introduction in three ways:
> * We now describe what is known about infant visual development, explaining the need for computational models that will help us to further understand how the brain learns and we highlight possible practical applications.
> * DNNs are already being extensively used as computational models for adult vision and in the second section of the introduction we present the relevant literature (Yamins and DiCarlo, 2016).
> * We have clarified how the DNNs’ learning process represents a promising tool to understand macro-scale characteristics of infants’ visual development.
>
> Moreover, we address your doubts about the parallel between DNNs training process and human learning in the following ways:
> * Although backpropagation is not biologically plausible, it can be approximated by a number of biologically plausible mechanisms, including the synaptic plasticity observed in top-down processes. The value of DNNs as models of the brain has also been recently discussed in two high-profile reviews (Richards et al., 2019; Sinz et al., 2019).
> * The parallel we intended to draw was unclear. DNNs are currently the best models of the adult visual system, but they do not attempt to map artificial neurons onto biological ones. Instead, activation profiles across the layer of a DNN show macroscale similarities to activation patterns in a brain region. We propose that similar macroscale statistics might provide insight into learning in the infant brain. This has been clarified throughout.
> * Infants’ access to labels is extremely impoverished. However, there is evidence that by 3-4 months of age, they can cluster together visually similar stimuli (Quinn, 1993). DeepCluster was therefore chosen as it is a state of the art unsupervised DNN which relies on a clustering strategy. Again, it was not our intention to claim a parallel between the specifics of the network and the brain, but rather that the DNN might capture some of the macroscale properties of the brain.
> * We agree that batch normalisation is not biologically plausible, or even applicable to online learning in machines. We have highlighted this in the discussion.
>
> Finally, we clarified the roles of genetic coding, learning, and random weight initialization:
> * The overall architecture of the brain is strongly shaped by genetics. However, the genetic code is far too limited to encode even a small proportion of the synaptic weights in the brain, and so a significant proportion of visual knowledge must be learned. This is why we learn to recognise object categories that are too recent to be specified in the genetic code, such as Pokémon, as highlighted by a recent paper (Janini and Konkle, 2019). Indeed, during the early stages of brain development, a large excess of synaptic connections is created which are then be pruned, to leave only the important ones. This may easily be thought of as a random initialisation followed by learning.
>
> We hope you agree that the paper is greatly strengthened.
>
> References
>   Janini D, Konkle T. A Pokémon-sized window into the human brain. Nature human behaviour. 2019 Jun;3(6):552.
>   Quinn PC, Eimas PD, Rosenkrantz SL. Evidence for representations of perceptually similar natural categories by 3-month-old and 4-month-old infants. Perception. 1993 Apr;22(4):463-75.
>   Richards BA, Lillicrap TP, Beaudoin P, Bengio Y, Bogacz R, Christensen A, Clopath C, Costa RP, de Berker A, Ganguli S, Gillon CJ. A deep learning framework for neuroscience. Nature neuroscience. 2019 Nov;22(11):1761-70.
>   Sinz FH, Pitkow X, Reimer J, Bethge M, Tolias AS. Engineering a less artificial intelligence. Neuron. 2019 Sep 25;103(6):967-79.
>   Yamins DL, DiCarlo JJ. Using goal-driven deep learning models to understand sensory cortex. Nature neuroscience. 2016 Mar;19(3):356.

---

> > ### Comment · AnonReviewer2 · 2019-11-14
> > **Response**
> >
> > Thank you for your careful and considered response.
> >
> > Regarding the parallel between DNN training and human learning, it seems to me that while there is a nontrivial body of work exploring the relationship between adult brains and fully-trained DNNs, there is not yet much work investigating whether the DNN training process should correlate with human learning.
> > As such, perhaps this work can serve as a stepping stone in this direction as the new abstract suggests.
> >
> > However, I am still skeptical about whether there is enough evidence to indicate that DNN training is likely to correlate with human learning. It may be that an existing DNN architecture correlates well with what we know about the human brain after the architecture is finished training, but that doesn't necessarily indicate that the same should be true during training. For DNN training itself, there are many variations such as different optimizers (SGD, SGD with momentum, Adam, etc) and other hyperparameters (gradient clipping, learning rate, etc); how should we pick these to best model human development?
> >
> > As for random weight initialization, I agree that a significant proportion of visual knowledge must be learned; however, the ability of babies to quickly perform tasks like looking at faces may indicate that there is at least some level of pre-encoded visual knowledge (e.g. edge detection, looking for areas of contrast, etc) in the human brain, which randomly initialized DNNs do not have. If babies do not need to learn such knowledge from scratch, but DNNs do, then it seems reasonable to expect that that could also lead to different outcomes during DNN training.
> >
> > Given my lack of experience in this area, I will defer to the other reviewers who have indicated greater experience.

---

> > > ### Author Response · Authors · 2019-11-15
> > > **An iterative approach**
> > >
> > > We appreciate your scepticism: this research was conceived and explicitly funded as a high-risk/high-gain programme. However, we believe the challenge is tractable because it can be tackled in an iterative way. Our current goal is to find initial points of contact between DNNs and infants, in the form of correspondences that are only loosely affected by the specific architecture, learning rule, optimiser, and hyperparameters. Once this initial contact is made, we will proceed to gradually more specific correspondences, iteratively refining our DNN design with the goal of converging on a more infant-like model. In the course of this process, we will aim to identify places where existing DNNs deviate strongly from infants. For example, if a subset of visual classes, such as faces or things that move, are learned much earlier by infants than any of the DNNs we test, this will suggest that perhaps an innate “face template” or “movement salience” is needed in the DNNs.
> > >
> > > Thank you for the time you have taken to engage with this work.

---

### Decision · Program_Chairs · 2019-12-19

**Decision:**

Reject

**Comment:**

This paper investigates the properties of deep neural networks as they learn, and how they may relate to human visual learning (e.g. how learning develops across regions of the infant brain). The paper received three reviews, all of which recommended Weak Reject. The reviewers generally felt the topic of the paper was very interesting, but overall felt that the insights that the paper revealed were relatively modest, and had concerns about the connections between DNN and human learning (e.g., the extent to which DNNs are biologically plausible -- including back propagation, batch normalization, random initialization, etc. -- and whether this matters for the conclusions of the present study). In response to comments, the authors undertook a significant revision to try to address these points of confusion. However, the reviewers were still skeptical and chose to keep their Weak Reject scores.

The AC agrees with reviewers that investigations of the similarity -- or not! -- between infant and deep neural networks is extremely interesting and, as the authors acknowledge, is a high risk but potentially very high reward research direction. However, in light of the reviews with unanimous Weak Reject decisions, the AC is not able to recommend acceptance at this time. I strongly encourage authors to continue this work and submit to another venue; this would seem to be a perfect match for CogSci conference, for example. We hope the reviews below help authors to improve their manuscript for this next submission.